# Parental Justifications for Not Vaccinating Children or Adolescents against Human Papillomavirus (HPV)

**DOI:** 10.3390/vaccines11030506

**Published:** 2023-02-22

**Authors:** Eliza S. Rodrigues, Elisa D. T. Mendes, Luciana B. Nucci

**Affiliations:** 1Health Sciences Post Graduate Program, School of Life Sciences, Pontifical Catholic University of Campinas, Campinas 13060-904, Brazil; 2Faculty of Medicine, Health Sciences Post Graduate Program, School of Life Sciences, Pontifical Catholic University of Campinas, Campinas 13060-904, Brazil

**Keywords:** vaccination refusal, human papillomavirus, HPV, papillomavirus vaccines, adolescent, health belief model

## Abstract

Vaccination coverage against Human Papillomavirus (HPV) is low compared with uptake of other vaccines in many countries, including Brazil. The aim of this study was to examine the main reasons provided by parents or guardians of a target population that did not have the first dose of HPV vaccine in a small rural Brazilian municipality, and to verify the factors associated with the reasons for non-vaccination. This is a cross-sectional study with interviews based on the Health Belief Model (HBM), conducted with parents and guardians of 177 unvaccinated children or adolescents. The outcome of interest was the main reason for not vaccinating the child/adolescent. The exposure factors of interest were knowledge about HPV and its prevention as well as sociodemographic characteristics. The main justifications for not vaccinating were lack of information (62.2%), fear or refusal (29.9%), and logistical issues (7.9%). The justifications associated with adolescents’ sex, fear, or refusal were mentioned by 39.3% (95% CI: 28.8–50.6%) of parents or guardians of girls and by 21.5% (95% CI: 13.7–31.2%) of parents or guardians of boys. The main barrier to HPV vaccination is lack of information. Further training of health professionals in clarifying the benefits of vaccination and differentiating the risks between boys and girls could encourage uptake.

## 1. Introduction

Human Papillomavirus (HPV) represents a group of viruses associated with sexually transmitted infections that are extremely prevalent worldwide [1]. Several studies have demonstrated the effectiveness of the vaccine in reducing the diagnosis of genital warts in both sexes [2] and cervical intraepithelial neoplasia grade 2+ in different age cohorts [3]. Nevertheless, HPV vaccination rates in 2014 were still very low, with significant differences noted between the more and less-developed regions, which had estimated coverage rates among the total female population of 7.1% and 0.6% for the first dose, respectively [4]. In 2020, the World Health Organization (WHO) published a global strategy to eliminate cervical cancer, which included a target of 90% of girls fully vaccinated against HPV by age 15 years by 2030 [5]. A modelling study in 78 low-income and lower-middle-income countries (LMICs) showed that girls-only HPV vaccination would lead to cervical cancer elimination in most LMICs if high coverage is reached (>90% coverage) given that the vaccine provides long-term protection [6]. In Latin America, which has a consistent history of high vaccination coverage with highly efficient national immunization programs, HPV vaccine uptake has been below expected [6,7,8]. Among 16 nations located to the south of the United States where HPV vaccination is offered, Mexico was the only country that met the target of 90% of girls fully vaccinated with the HPV vaccine by age 15 in 2020 [9]. In Brazil, data from a representative sample of female students, aged between 13 and 17 years, selected from among 6th–12th grade students at public and private schools in representative urban and rural areas of Brazil, showed HPV vaccination coverage of 48.9%, well below the recommended target of 90% to be achieved by 2030 [10].

Although essential, encouraging vaccination against HPV prompted an ethical and social discussion about offering protection against a sexually transmitted virus for very young children or adolescents [11]. Some authors identify difficulties in accessing services and information about HPV vaccines [12,13,14] in addition to perceptions and behaviors that influence refusal in a different social context [14]. This reveals the heterogeneity of vaccination scope in Brazil, namely, higher with the first dose in urbanized areas and in groups with better socioeconomic status [15].

In order to contribute to prevention strategies and address possible flaws in current campaigns, we studied the reasons why parents or guardians preferred not to vaccinate their children or adolescents against HPV and the associated factors in a small, less urbanized, and low-income municipality of Brazil.

## 2. Materials and Methods

### 2.1. Study Design and Sample

This is a cross-sectional study conducted in a small Brazilian municipality, Alterosa, in the state of Minas Gerais. The municipality has a resident population of 13,717 inhabitants, similar to 70.3% of the other 5565 Brazilian municipalities, according to 2010 Census data. It is an agricultural municipality with almost the entire population dependent on the Unified Health System (SUS), with an urbanization rate of 72.9%, an average monthly salary of 1.6 minimum wages, and 34.5% of the population with a monthly income of up to ½ the minimum wage [16].

The study population was defined as all domiciles of parents or guardians of children or adolescents who should have been immunized against HPV in 2018, according to the recommendations of the Brazilian Ministry of Health (in 2014: girls aged 11–13 years; and in 2017: girls aged 9–14 and boys aged 11–14). The total of unvaccinated children or adolescents was identified from lists provided by the Municipal Health Department and the Coordination of Epidemiological Surveillance. The merging of these two lists, together with exclusions and losses, is shown in Figure 1. Briefly, unvaccinated children or adolescents (n = 1338) were identified from a list provided by the Municipal Health Department of the entire population of the municipality registered in the Family Health Strategy (FHS). The FHS is a Brazilian community-oriented primary healthcare model with practices focused on comprehensiveness and health promotion. The program has a particular focus on reaching the most vulnerable population [17]. Through the Coordination of Epidemiological Surveillance, another list was provided of individuals who had received the first dose of the HPV vaccine (n = 638). After merging these two lists (n = 900), an active search was performed excluding 449 people who were already vaccinated, 45 because data of another child/adolescent in the household were taken, and 7 who were older than the target population. From the remaining sample (n = 399), 91 no longer resided in the city, 99 were not found at home after 3 visits on different days of week at different times, 22 refused to answer, and 5 said they were vaccinated, but did not have any record of this, resulting in a study sample of 182 parents or guardians of unvaccinated children or adolescents.

After signing the informed consent form, parents or guardians of unvaccinated children or adolescents were invited to participate in the interview. Data collection was carried out in person through home interviews and based on a questionnaire. The questionnaire was designed according to the HBM [18], with the interviews conducted between January and October 2019. Interviewers were trained by the lead researcher and were given a manual for conducting the interviews as well as standardized guidelines on the questionnaire. If there was more than one child/adolescent who had not been vaccinated at the same address, a mobile application (Randomizer) was used to decide which child/adolescent would be the subject of the interview in that domicile.

### 2.2. Measures

The outcome of interest was measured using a specific question, which asked about the main reason for not vaccinating the child/adolescent, to which the respondent’s spontaneous answer was expected. According to the HBM [18], the final 16 reasons were classified as: 1. Lack of information, where the interviewee reported not receiving guidance from a health professional that he/she did not know about the vaccine, or that the teenager was very young or was not sexually active; 2. Fear or refusal, where the interviewee reported (a) fear of the injection, (b) fear of side effects, (c) mentioned news or social network coverage, (d) that their religion prohibited vaccination, or (e) that the teenager refused or did not believe in the vaccine; and 3. Logistical issues, such as forgetting appointments, lack of time to attend vaccination, restricted vaccination room hours, or a shortage of vaccine in the unit [19,20,21]. To define the main reason for non-adherence, the following hierarchical criteria were applied: 1. Lack of information, 2. Fear or refusal, and 3. Logistical issues. In 177 (97.3%) respondents, it was possible to define one main reason for non-adherence according to the hierarchy described in the methods. Of the remaining five respondents, four said that they did not know the reason and one said that he lived in another state at the time when the child/adolescent should have been vaccinated.

The exposure factors of interest were sex and age of the adolescent; age of the interviewee; kinship with the adolescent; social class, classified into A, B1, B2, C1, C2, D–E; and religion of the family. Knowledge about HPV and prevention was assessed on a Likert scale, with scores ranging from one to five, derived from five questions (i.e., three about knowledge and two about prevention). A Likert scale ranging from “I totally disagree” to “I totally agree,” or the option of “Don’t know/Didn’t want to answer” was used, with higher values for the correct answers and a zero value for “Don’t know/Didn’t want to answer.” Four domains of the HBM [18] were also evaluated: 1. perceived benefits of vaccines and of the HPV vaccine, 2. perceived barriers to vaccination and to HPV vaccination, 3. perceived vulnerability, and 4. perceived severity (Table 1). Responses were again based on a Likert scale, ranging from “I totally disagree” to “I totally agree,” or the option of “Don’t know/Didn’t want to answer.” Values from one to five were assigned to define the scores for the HBM domains, with higher values for the responses that indicated the intention to vaccinate. A zero value was assigned to the option “Don’t know/Didn’t want to answer.”

### 2.3. Analysis

Descriptive statistical analyses were presented as mean and 95% confidence intervals (95% CI) for the assessed scores (quantitative variables), and absolute and relative frequencies for categorical variables. Analyses to verify the assumptions of normality of the data were also performed. The association between justifications for non-vaccination and sociodemographic characteristics of the interviewee, family and child/adolescent were verified using the Chi-square statistical test or Fisher’s exact test for categorical variables and the F-test (ANOVA-Analysis of Variance) for knowledge scores on HPV and prevention and domains of the HBM. All analyses were performed using the SAS on Demand for Academics (SAS Studio version 3.8) statistical package. A significance level of 0.05 (α) was adopted.

## 3. Results

The study sample consisted of 182 parents or guardians of children or adolescents who did not take the first dose of the HPV vaccine. The description of the reported reasons considered all the interviewees’ answers, including cases in which they spontaneously reported more than one reason (n = 49). Among the main reasons cited for not vaccinating against HPV, 51.7% (n = 94) of respondents answered that they were not advised by health professionals, 15.9% (n = 29) reported fear of the injection, while forgetfulness was reported by 6.6% (n = 12) of respondents as a reason for not vaccinating (Table 2).

Lack of information was the reason most frequently reported by parents (62%; n = 110), followed by fear or refusal (30%; n = 53) and those who were willing to have the vaccination but were unable to do so for logistical reasons (8%; n = 14) (Figure 2).

Among the respondents with valid justification (n = 177), 39.0% (n = 69) were 41–49 years old, 92.7% (n = 164) were female, 89.3% (n = 158) were fathers/mothers of the child/adolescent, 70.3% (n = 123) were Catholics, and 47.7% (n = 82) belonged to social class C. There was no statistically significant difference in relation to parent/guardian age, sex, or kinship with the children, nor with the family’s religion, social class, or the adolescent’s age. Parents or guardians of boys were more likely to cite lack of information (68.8%) and logistical issues (9.7%) than parents or guardians of girls (54.8% and 6.0%), respectively). Refusal or fear was reported more frequently by parents or guardians of girls (39.3%) than by parents or guardians of boys (21.5%) (Table 3).

Knowledge of HPV prevention was higher for those who cited a lack of information or fear/refusal as justification compared with those who did not vaccinate for logistical reasons, but without statistical significance. Regarding the health belief model, the benefits of taking the vaccine and perceived vulnerability were the highest-scoring domains in the three groups, indicating a greater tendency to vaccination. There were no statistically significant differences between the reasons for non-vaccination in any of the items evaluated (Table 4).

## 4. Discussion

This study evaluated data on parents or guardians of children or adolescents living in a small and rural town in the interior of Brazil, who did not vaccinate their children or adolescents against HPV. The data included sociodemographic characteristics, knowledge of HPV and vaccination, and justification for non-vaccination. The main reasons cited by the interviewees for not vaccinating their children were lack of information, followed by fear or refusal due to fear of side effects, and news stories seen on social media, as reported by others studies [14,20]. Using the same HBM method, other studies also suggest that knowledge of this subject is insufficient, influencing lower perception of HPV vaccine benefits as one of the main reason for not vaccinating [12,14,22,23]. Better awareness of HPV-transmitted infections and the importance of vaccination by parents and guardians, as well as children and adolescents, may increase the acceptability of vaccination [24].

Forgetting appointments and access difficulties, classified as “logistical issues,” were the third most frequent reason. The sex of the child/adolescent was associated with the justification for not vaccinating, with fear or refusal being cited more frequently by parents or guardians of girls than boys.

A study of different age cohorts of Brazilian girls showed heterogeneous coverage in micro regions, higher with the first dose, in urbanized areas and in groups with higher socioeconomic status [15]. In rural areas in the interior of the country, logistical barriers and lack of information about vaccine benefits can override refusal.

We found that the child/adolescent’s gender is related to the justification given by parents or guardians, with fear or refusal being more frequently cited by parents of girls (38.4%) than parents of boys (20.4%). This higher percentage of refusal for girls may be explained by the earlier recommended age for starting vaccination and cultural issues related to sexuality. A similar study published in Sweden found that parents who showed resistance to allowing HPV vaccination for their daughter at her current age were willing to wait until she was older [25].

Regarding vulnerability according to the HBM, approximately 30% agreed that the child/adolescent was too young to think about a sexually transmitted disease. Our results are similar to the study of Krawczyk (2015), which showed that a small portion of respondents questioned the real severity of HPV infection, while a larger number responded that their daughters were not susceptible to such infection [26]. Likewise, the association of threat assessment (operationalized as perceived vulnerability of the child to HPV and perceived severity of HPV) with the intention to vaccinate was not directly or indirectly confirmed in the study of Olagoke (2021) [27]. However, Frio and França (2020) found no association between vaccination and initiation of sexual activity in girls in a National Survey of School Health in Brazil [28].

Health professionals involved in vaccination need to receive reliable information about the HPV vaccine, preferably from trusted medical institutions or official organizations, to increase the probability of recommending the vaccination with certainty. Effective training of health professionals and a comprehensive protocol for providers to address hesitant parents could contribute to a more efficient recommendation, increasing their influence on the patient’s acceptance or those responsible for receiving the HPV vaccination [29,30]. Training of the health team on information and clarification about the vaccine and its benefits could have a relevant impact on the Brazilian National Health System [31]. Awareness about sexually transmitted infections in primary health care is fundamental, reinforcing the recommendation for vaccination in the right age range [32].

The Brazilian National Immunization Program (PNI) was created in 1973 by the Brazilian Ministry of Health and is recognized for promoting free vaccination against more than 15 immunogens. PNI has reported successful results in recent decades, with results and outcomes similar to those in developed countries, and a positive impact on mortality from infectious diseases. Despite this, there has been an important drop in vaccination coverage rates in Brazil since 2016. This is due to multifactorial causes, which include problems with access to the vaccine, but with an important component of vaccine hesitancy. In European and North American countries, vaccine hesitancy is an established concern, but in Brazil, it is still a situation that is rarely evaluated in the literature, despite being well established, especially after the COVID-19 pandemic [33]. The Institute for Health Policy Studies (IEPS) published a report on the Panorama of Vaccination Coverage in Brazil in 2020, based on data from the Information System of the National Immunization Program (SI-PNI). This was available through the TABNET portal, from the Department of Informatics of the SUS (DATASUS) [34]. This report demonstrates a universal decline in recent vaccination coverage rates in Brazil in recent years. In 2015, 71.3% of Brazilian municipalities were above the immunization target, but this percentage dropped to 46.2% in 2020. Despite this drop, according to data from the information system, the vaccination coverage rate in the municipality evaluated in the year 2018 was 77.13%, which is much higher than the evaluation for the HPV vaccine.

Despite the official low percentage of vaccination against HPV in the municipality, the unvaccinated target audience may be overestimated, since we found the percentage of vaccinated children or adolescents to be much higher than expected. Therefore, a limitation of the study was the use of secondary data to locate respondents and, as the aim was to assess the reasons for non-vaccination against HPV, a census of vaccinated and non-vaccinated persons was not carried out. Another limitation of the study refers to possible recall bias, since we were investigating the reasons for a decision that was made in the past. However, the questionnaire was structured to introduce the theme and stimulate memory on the subject, although it was not validated. To the best of our knowledge, when the project for this study was developed, there was no validated questionnaire for this purpose. In 2018, a comprehensive study with validation of a questionnaire was found; however, this questionnaire was aimed at adolescents and young people aged 16 to 25 years [32,35]. It should also be noted that as the economy of the municipality is based on agriculture, the immigration rate rises during the harvest season, which explains the percentage of people (22.4%; n = 91) who no longer resided in the municipality. Although the percentage of people who no longer resided in the city and those who were not found totaled 47.6%, possible selection bias is within the acceptable range for a cross-sectional study with an active search [36].

The HBM has been applied in several regions of the world, allowing comparisons [22,23,27,37,38,39,40]. It provides information on behaviors allowing to list interventions that focus on individuals’ attitudes and beliefs to overcome low vaccination coverage.

## 5. Conclusions

In conclusion, we found lack of information to be the main reason for non-vaccination in the small, rural town under study. These circumstances may be similar in other municipalities with similar contexts. The association of the sex of the child/adolescent with the reason for non-vaccination indicates that information must be provided to dissociate the vaccine against HPV from the beginning of sexual activity. Since the recommended age for the vaccination of girls is lower than that for boys, the risks of HPV for both must be clarified. In addition, efforts are urgently required to promote the training of health professionals as sources of information on HPV and vaccination in order to encourage parents or guardians to vaccinate children or adolescents. With better provision of information, the target audience might be reached more effectively as part of health campaigns for HPV vaccination.

## Figures and Tables

**Figure 1 vaccines-11-00506-f001:**
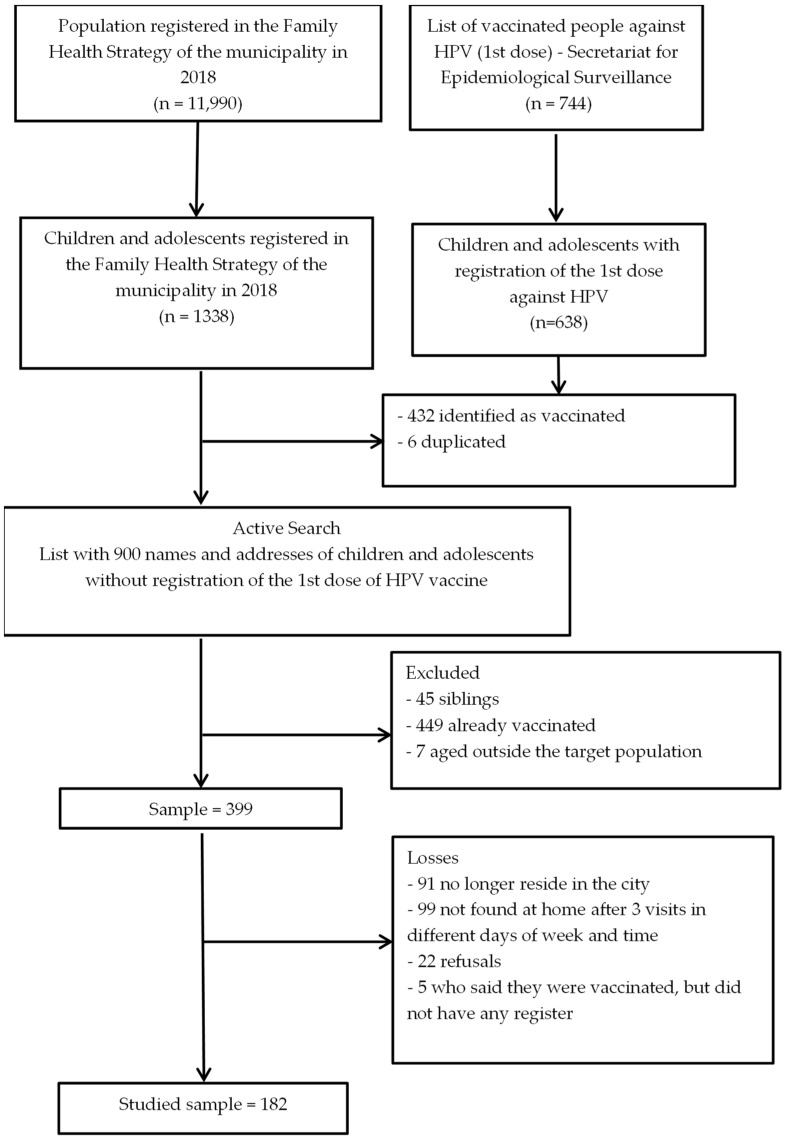
Flowchart of the sample selection process.

**Figure 2 vaccines-11-00506-f002:**
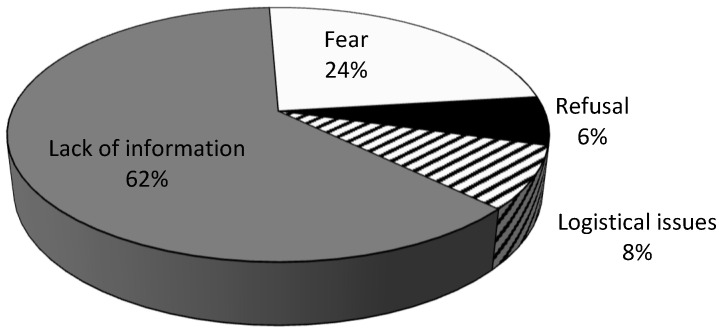
Main reason reported by parents or guardians for not vaccinating their children or adolescents against HPV (n = 177).

**Table 1 vaccines-11-00506-t001:** Categories and questions in the Health Belief Model.

Category	Questions
Perceived benefits of vaccines	I have no confidence in vaccines
	Vaccines are not effective in preventing diseases
	It is not necessary to take all vaccines
	It is preferable to contract the disease to obtain natural protection
Perceived benefits of the HPV vaccine	I do not consider the HPV vaccine safe
	I believe that if I receive the HPV vaccine, I will not be protected against cervical cancer
	I believe that if I receive the HPV vaccine, I will not be protected against HPV
	The HPV vaccine prevents the appearance of genital warts (condyloma)
	The HPV vaccine prevents cervical cancer
Perceived barriers to vaccination	I don’t have enough information about vaccine-preventable infections
	I don’t have enough information about vaccines
	The vaccination unit is difficult to access
Perceived barriers to HPV vaccination	The pediatrician/general practitioner discouraged me from taking <the teenager> to get the HPV vaccine
	Social media / internet discouraged me from getting the HPV vaccine
	The HPV vaccine has severe side effects
	It is difficult to find the vaccine at the clinic
Perceived vulnerability	I think the <adolescent> is too young to think about sexually transmitted diseases
	Women can be infected with HPV
	Men can be infected with HPV
Perceived severity	HPV can cause a serious illness
	Cervical cancer can be dangerous and lead to death
	HPV can cause other types of cancer

**Table 2 vaccines-11-00506-t002:** Reasons for non-vaccination against HPV by parents or guardians of unvaccinated children and adolescents (n = 182).

Categories	Reasons for Non-Vaccination	n	%
Lack of information	Not receiving guidance from a health professional	94	51.7
	Did not know about the vaccine	39	21.4
	Child/adolescent is very young	3	1.7
	Child/adolescent is not sexually active	3	1.7
Fear or refusal	Fear of injection	29	15.9
	Fear of side effects	15	8.2
	Mentioned news or social network coverage	10	5.5
	Child/adolescent refused	8	4.4
	I don’t believe in this vaccine	6	3.3
	Religion prohibited vaccination	2	1.1
Logistic issues	Forgetfulness	12	6.6
	Lack of time	8	4.4
	Tried, but vaccination unit was closed	7	3.9
	Shortage of vaccine	3	1.7
	Restricted vaccination hours	1	0.6
	Other reasons	8	4.4

**Table 3 vaccines-11-00506-t003:** Sociodemographic characteristics of parents or guardians of children and adolescents not vaccinated against HPV; total sample, according to main reason for non-vaccination (n = 177).

	Total Sample	Lack of Information (n = 110)	Fear or Refusal (n = 53)	Logistical Issues (n = 14)	
Characteristics	n (%) *	n (%) **	n (%) **	n (%) **	*p*-Value
Age (in years)					
≤40	66 (37.3)	35 (53.0)	24 (36.4)	7 (10.6)	0.293 ^a^
41 to 49	69 (39.0)	44 (63.8)	20 (29.0)	5 (7.3)	
≥50	42 (23.7)	31 (73.8)	9 (21.4)	2 (7.8)	
Sex					
Male	13 (7.3)	8 (61.5)	4 (30.8)	1 (7.7)	1.000 ^b^
Female	164 (92.7)	102 (62.2)	49 (29.9)	13 (7.9)	
Relationship with the child/adolescent					
Father/mother	158 (89.3)	98 (62.0)	46 (29.1)	14 (8.9)	0.363 ^a^
Other	19 (10.7)	12 (63.2)	7 (36.8)	0 (0.0)	
Religion (n = 175)					
Catholic	123 (70.3)	77 (62.6)	37 (30.1)	9 (7.3)	0.630 ^b^
Christian	42 (24.0)	25 (59.5)	14 (33.3)	3 (7.1)	
Other	10 (5.7)	6 (60.0)	2 (20.0)	2 (20.0)	
Social status (n = 172)					
A–B	46 (26.7)	32 (69.6)	12 (26.1)	2 (4.4)	0.608 ^b^
C	82 (47.7)	50 (61.0)	24 (29.3)	8 (9.8)	
D–E	44 (25.6)	24 (54.6)	16 (36.4)	4 (9.1)	
Children/adolescent					
age (in years)					
≤15	71 (40.1)	39 (54.9)	24 (33.8)	8 (11.3)	0.424 ^a^
16–17	70 (39.5)	47 (67.1)	20 (28.6)	3 (4.3)	
≥18	36 (20.3)	24 (66.7)	9 (25.0)	3 (8.3)	
Sex					
Male	93 (52.5)	64 (68.8)	20 (21.5)	9 (9.7)	0.033 ^a^
Female	84 (47.5)	46 (54.8)	33 (39.3)	5 (6.0)	

* percentage refers to total sample; ^a^: Chi-square test. ** percentage refers to the total of the category; ^b^: Fisher’s exact test.

**Table 4 vaccines-11-00506-t004:** Main reasons for non-vaccination according to knowledge scores about the Human Papillomavirus (HPV) and about prevention and domains of the health belief model (n = 177).

Domains	Lack of Information	Fear or Refusal	Logistical Issues	*p*-Value
	Mean (95%CI)	Mean (95%CI)	Mean (95%CI)	
Knowledge about HPV	3.33 (3.17–3.49)	3.25 (3.00–3.50)	3.05 (2.35–3.74)	0.503
Knowledge about prevention	4.18 (4.04–4.31)	4.19 (3.97–4.41)	3.79 (2.94–4.63)	0.221
Perceived benefits of vaccines	4.06 (3.92–4.20)	3.97 (3.68–4.25)	4.14 (3.36–4.92)	0.746
Perceived benefits of HPV vaccine	3.62 (3.43–3.81)	3.58 (3.32–3.85)	3.82 (3.21–4.42)	0.752
Perceived barriers of vaccines	2.98 (2.83–3.12)	3.06 (2.81–3.31)	2.92 (2.38–3.46)	0.771
Perceived barriers of HPV vaccines	3.44 (3.29–3.59)	3.36 (3.10–3.62)	3.21 (2.48–3.95)	0.635
Perceived vulnerability	3.81 (3.67–3.95)	3.84 (3.60–4.07)	4.31 (3.94–4.68)	0.094
Perceived gravity	3.72 (3.56–3.89)	3.77 (3.53–4.01)	4.05 (3.63–4.48)	0.422

CI: Confidence Interval.

## Data Availability

The data will be available upon reasonable request to the corresponding authors.

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
