# Peer review of "Parental Justifications for Not Vaccinating Children or Adolescents against Human Papillomavirus (HPV)"

_vaccines, 2023, doi:10.3390/vaccines11030506_

Round 1

Reviewer 1 Report

The authors have conducted a survey/study to find parental justifications for not vaccinating children or adolescents against HPV. The study is well designed and written, yet it is not a novelty. The study was carried out in a small rural Brazilian area and thus the results were expected. Why this particular area? It does not represent the overall population and it would be interesting to compare with other small rural areas as well as with bigger urban places.

Do the authors (or others) have conducted the same study on other vaccines? Is the success rate of other vaccines different and if so, are parents more informed about them? Is HPV vaccine free in Brazil?

Author Response

  1. Why this particular area? It does not represent the overall population and it would be interesting to compare with other small rural areas as well as with bigger urban places.

We agreed that the comparison with other rural and urban areas would be interesting, to evaluate the possible particularities of the chosen area. However, the municipality was chosen precisely because it is similar to 70.3% of the 5,565 Brazilian municipalities, according to data from the last Census (2010), therefore considered a sample of the primary care policy provided by the single municipal health system in our country. This data is described in the results.

  1. Do the authors (or others) have conducted the same study on other vaccines? Is the success rate of other vaccines different and if so, are parents more informed about them? Is HPV vaccine free in Brazil?

We appreciated the comments and agreed that it would be interesting to evaluate the adherence rate of other vaccines in the evaluated municipality. However, unfortunately, this data was not available, official vaccine coverage data do not include the HPV vaccine and specific vaccines for each age group are not available. For the target population (girls aged 9-14y and boys aged 11-14y) HPV vaccine is free. Although, we thought it interesting to include a paragraph in the discussion section: “The Brazilian National Immunization Program (PNI) was created in 1973 by the Brazilian Ministry of Health and is recognized for promoting free vaccination of more than 15 immunogens. PNI presents successful results in recent decades, with results and outcomes similar to developed countries, with a positive impact on mortality from infectious diseases. Despite this, there has been an important drop in vaccination coverage rates in Brazil since 2016, due to multifactorial causes, which include problems with access to the vaccine, but with an important component of vaccine hesitancy. In European and North American countries, vaccine hesitancy is an established concern, but in Brazil, it is still a situation rarely evaluated in the literature, despite being well-established, especially after the Covid-19 pandemic. (New ref: Sato APS, 2018).The Institute for Health Policy Studies (IEPS) published a report on the Panorama of Vaccination Coverage in Brazil, 2020 (New ref: https://ieps.org.br/wp-content/uploads/2021/05/Panorama_IEPS_01.pdf ) based on data from the Information System of the National Immunization Program (SI-PNI), available through the TABNET portal, from the Department of Informatics of the SUS (DATASUS). This report demonstrates a universal decline in recent vaccination coverage rates in Brazil in recent years. In 2015, 71.3% of Brazilian municipalities were above the immunization target, as this percentage dropped to 46.2% in 2020. Despite this drop, according to data from the information system, the vaccination coverage rate in the municipality evaluated in the year 2018 was 77.13%, much higher than the evaluation for the HPV vaccine.”

Reviewer 2 Report

The authors want to analyze the factors that are related to non-vaccination against HPV in the adolescent population of a specific area of ​​Brazil. For this objective they carried out a transversal study composed by 172 or 182 subjects (in abstracts n=172 y the text 182). May main concern about this study, I do not know, if the number of populations included is enough to get it. In the materials of methods section does the simple size is not included, nor is it justified that with those 172 or 182 subjects (it is not clear) the main objective is achieved.

In the other hand, it is a little new or original study. 

Author Response

We appreciate the commentary. The study sample consisted of 182 parents/guardians of children/adolescents who did not take the first dose of the HPV vaccine. In 177 of the 182 respondents, it was possible to define one main reason for non-adherence according to the hierarchy described in the methods. Of the remaining five, four said that they did not know the reason and one said that he lived in another state at the time when the child/adolescent should have been vaccinated. In conclusion, we had 182 responders and 177 respondents with valid justification.